# Understanding the Interplay between Antimicrobial Resistance, Microplastics and Xenobiotic Contaminants: A Leap towards One Health?

**DOI:** 10.3390/ijerph20010042

**Published:** 2022-12-20

**Authors:** Federica Piergiacomo, Lorenzo Brusetti, Leonardo Pagani

**Affiliations:** 1Faculty of Science and Technology, Free University of Bolzano-Bozen, Piazza Università 1, 39100 Bolzano, Italy; 2Antimicrobial Stewardship Project, Provincial Hospital of Bolzano (SABES-ASDAA), Lehrkrankenhaus der Paracelsus Medizinischen Privatuniversität, 39100 Bolzano, Italy

**Keywords:** microplastics, One Health, antimicrobial resistance, xenobiotic contaminants, environmental threatening, global health safety

## Abstract

According to the World Health Organization, the two major public health threats in the twenty-first century are antibiotic-resistant bacteria and antibiotic-resistant genes. The reason for the global prevalence and the constant increase of antibiotic-resistant bacteria is owed to the steady rise in overall antimicrobial consumption in several medical, domestic, agricultural, industrial, and veterinary applications, with consequent environmental release. These antibiotic residues may directly contaminate terrestrial and aquatic environments in which antibiotic-resistance genes are also present. Reports suggest that metal contamination is one of the main drivers of antimicrobial resistance (AMR). Moreover, the abundance of antibiotic-resistance genes is directly connected to the predominance of metal concentrations in the environment. In addition, microplastics have become a threat as emerging contaminants because of their ubiquitous presence, bio-inertness, toughness, danger to aquatic life, and human health implications. In the environment, microplastics and AMR are interconnected through biofilms, where genetic information (e.g., ARGs) is horizontally transferred between bacteria. From this perspective, we tried to summarize what is currently known on this topic and to propose a more effective One Health policy to tackle these threats.

## 1. Introduction

### 1.1. State of the Art

The biological and physicochemical properties of the environment have been adversely affected by rising energy and freshwater demand, and intensive agriculture and farming are tightly intertwined with population growth, urbanization, and industrialization. Hence, the environment itself has become more susceptible to phenomena such as antibiotic resistance (AMR), an acquired trait that makes microorganisms able to survive, grow, and reproduce even in the presence of significant concentrations of antibiotics [1,2]. According to a report by Ram and Kumar from the World Health Organization, in our century, the two most important issues in public health are antibiotic-resistant genes (ARGs) and, consequently, antibiotic-resistant pathogenic bacteria (ARB) [3]. The reason for the global prevalence and the constant increase of ARB is owed to the steady rise in overall consumption (and consequent environmental release) of antibiotics in several medical, domestic, agricultural, industrial, and veterinary applications; the implication of ARB is today a substantial dread in managing human- or animal-related severe illnesses [4,5]. The outdoor environments can be easily polluted by several kinds of antibiotic residues derived from civil wastewater, manure, hospital waste, or even from pharmaceutical industry pollution. Moreover, their residues in soils are also enhanced by many agricultural practices, such as the use of manure and sewage sludge [6]. Such antibiotic misuse has dramatically influenced the environmental microbial ecology, and the abundance of ARB and ARG in different environments, such as in urban wastewater, freshwater sediments, municipal solid-waste leachate, husbandry sludge, freshwater/drinking water and groundwater [3,7,8,9,10,11].

Urban discharge into inland waters and the prevalence and diversity of ARB have been significantly correlated, as shown in several studies [12,13]. A positive correlation has indeed been demonstrated between the concentration of some antibiotics and their corresponding ARGs in inland waters, including wastewater treatment plants (WWTPs) [14]. It has also been reported that in WWTPs, the effluent displayed more ARB in percentage than the influent due to various factors, such as enhanced antimicrobial activity, water treatment conditions, chemical properties of antimicrobial agents, and bacterial genome dynamics, including gene transfer [3,15]. The overall data showed that WWTPs are not efficient in removing antibiotics, making them potential spots for gene transfer. The reasons why WWTP may increase the number of ARB in the effluent include the general bioavailability of antibiotics [7], the high bacterial density in the biofilms [16], and their augmented nutritional status.

#### 1.1.1. Metal Contamination and ARG Spread 

The presence and distribution of ARGs are also significantly correlated with other anthropogenic pollutants [14]. These include heavy metal ions, organometallic molecules, disinfectants, surfactants, biocides, and chemical solvents [17,18]. In addition, AMR is further spread by co-selection mechanisms that favor the dispersal of mobile genetic elements [19,20,21,22]. 

Boundless discharge of heavy metals into the environment can induce serious losses and dangers in microbial communities, organisms, and humans. Heavy metals, both from natural and anthropogenic sources [23,24], steadily bioaccumulate in living organisms, moving through the food chain from the bottom to the top trophic level. Unlike antibiotics, they persist for a very long time due to their non-degradability, and then, when they reach threshold concentrations, they start exerting toxicity. Metal elements involved in the biological organism’s formation and growth, such as Fe, Cu, and Zn, become toxic only if present in high concentrations; toxicity at lower concentrations can instead be exhibited by heavy metals, such as Hg, Pb, and Cd. Under certain circumstances, they could also form metallic complexes, boosting the toxicological implications of their presence [25,26,27]. 

Natural microbial communities have adapted their morphological structures and physiological and biochemical properties to develop and improve tolerance to heavy metals through evolutionary mechanisms. These mechanisms were reported from studies in the environment and rely on intracellular bioaccumulation, extracellular sequestration, redox reaction of heavy metals, bio-precipitation, and efflux pump systems [24].

Interestingly, a huge number of reports suggest that metal contamination is one of the main drivers of resistance against almost all antibiotics currently used in medical practices, provoking the transfer of ARGs and multidrug resistance among bacteria through selection and co-selection pressures [24,28,29,30,31], and through point mutations or changes at the genetic level and expression [32]. In many circumstances, the abundance of AMR genes is directly connected to the predominance of heavy metal concentrations in the environment [33,34]. Stepanauskas et al. [30] showed that nickel (Ni) and cadmium (Cd) predominant contamination increased bacterial resistance to ampicillin or chloramphenicol. Likewise, the combined co-selection of resistance to tetracycline, ampicillin, and chloramphenicol was related to the selection of copper (Cu) resistance due to agricultural soils contaminated with Cu [35]. 

Research has also shown that bacterial selection generally occurs through co-resistance, cross-resistance, and co-regulation [19]. The mechanism of co-resistance refers to the fact that several antigenic determinants, which are equally resistant to different toxic components, are localized in the same transposable genetic element. Cross-resistance means that the same removable genetic element brings the same antigenic determinant that can generate resistance to both antibiotics and heavy metals. Co-regulation grants a coordinated backlash to several toxic compound expositions [36].

Metallic nanoparticles (NPs-metals and metal oxides) have bactericidal synergistic effects with antibiotics [37,38]. Because of their high reactiveness and targeting action of microbial cells, NP-based antibacterial products have been widely employed in medical, food, and cosmetic contexts, as well as in wastewater treatments (e.g., membrane filtering methods) with the purpose of controlling infections. After many years, this huge chronic bacterial exposure to NP reagents has also raised the risk of microbes becoming increasingly tolerant. Hence, it is becoming more evident that resistance in bacteria could be promoted by the constant presence of NP sub-lethal or sub-inhibitory levels. Recent studies have shown that progressively growing concentrations of Cu and silver NPs can be tolerated by bacteria [39,40,41,42]. Indeed, NPs can enhance the horizontal gene transfer (HGT) of ARGs, thus fostering conjugation and transformation processes between bacteria in multiple environments (e.g., laboratory cultures [43,44], natural environments [45,46], and anthropological systems [47]). The same plasmid could harbor genes encoding silver resistance and antibiotic resistance, and integrons may also ease ARG co-selection. Moreover, bacteria seem to upregulate efflux pumps to expel metal ions together with antibiotics, acquiring cross-resistance. Ultimately, Zhang et al. [48] uncovered nanoalumina- and ZnO-NP-induced mutations in the *gyrA* and *soxR* genes, conferring resistance to ciprofloxacin and chloramphenicol in *Escherichia coli*. This means that NPs could also promote AR by fostering mutations [37,49].

#### 1.1.2. Microplastic Contamination and ARG Spread

In many natural ecosystems, the phenomenon of ARG spread can be amplified by the presence of other pollutants, such as microplastics, in the inhabitant microbiota [19].

Microplastics (MPs: any plastic fragment smaller than 5 mm in size) have become a threat as an emerging contaminant, reaching 359 million tons worldwide in 2018 [50]. To reduce their impact on the environment, several public actions have been executed, including the UE initiative to ban the use of microplastics in European toiletries (UNEP40). The main reasons for global concern are the durability and general inertness of their chemical bonds, their pervasive presence in all the environments, and danger to biotic life, most likely including human health [3]. Moreover, complete plastic mineralization requires hundreds to thousands of years, meaning a long-term life and contaminant presence [3,51]. Finally, microplastic surfaces can adsorb and bond not only pollutants such as heavy metals, thus fostering cross-resistance [24], but also viruses, microorganisms, and complex molecules [51]. WWTPs have not been designed to remove microplastics, and from a median-sized plant (average treatment capacity of 5 × 10^7^ m^3^/year), a daily discharge of up to 2 million plastic microparticles is estimated [52]. Most of the studies have explored the effects of microplastics’ presence on the communities’ response for the performance of the plant, such as on the removal of nitrogen toxic compounds in activated sludges [53,54,55], while fewer are available on the interactions between antibiotics and microbes inside the biofilms attached to microplastics in the plant [55]. Accordingly, Pham et al. [55] determined that microplastics from polyethylene (PE) and polystyrene (PS) could enhance the proliferation of sulfonamide-enriched biofilms in domestic WWTP-activated sludge samples.

From the WWTPs, microplastics can easily reach the outflowing canal and be dispersed in the receiving water body [56], including the river water column [57] and its sediments [58,59,60]; from there, they are going to reach marine water, beaches, and be dispersed into indoor air [61], different types of soil [62,63,64], and river and coastal sediments [65,66], fauna biota (mussels and fish) [67], and even in human blood, where polyethylene terephthalate, polyethylene, polymers of styrene and methyl-methacrylate were identified for the first time [68].

Urban and agricultural soils, and consequently vegetables and livestock, are especially assumed to be vulnerable to microplastic contamination: potential incoming pathways are littering (including debris from plastic mulch and greenhouse plastic covering), street runoff (including tire wear, atmospheric deposition, plastic mulching), irrigation with freshwater or wastewater, use of organic fertilizers derived from bio-waste, sewage sludge, or manure [69,70,71].

Ecosystems and their health can be clearly undermined by microplastics through several mechanisms: microplastics may induce changes in organism population balance, and any deleterious effect on a single species could indeed have overlooked consequences in the ecosystem. Indeed, microalgae populations can both be harmed if microplastics hinder the absorption of essential nutrients, and benefit if these reduce populations of primary consumers [72]. Conversely, in the clam *Atactodea striata*, energy uptake is affected and consequently influences energy transfer in the food web [73]. Likewise, the taxonomic abundance, richness, and diversity of the benthic fauna were found to have been increased by plastic debris, leading to a significant change in the community structure [74]. In the same way, microbial communities colonizing microplastics, the plastisphere, present different functional properties, compositions, and structures, with a potential ecological impact on the overall ecosystem biogeochemistry [75,76]. In this context, the microbiota on the microplastic surfaces could have different ecological functionalities; the ultimate consequence of such a shift is still partially unknown and context-dependent since they vary among different environments [77]. 

Bacteria can use the microplastic surface as a substrate for biofilm formation—where microbial cells are highly concentrated and embedded in extracellular polymers—sustaining their dispersal to new regions [78]. In this context, cells are facilitated in nutrient or metabolite exchanges, syntrophic behavior, cellular communication, self-protection, and stress resistance through enhanced HGT [79]. The high resistance and usually low density of MPs provide ideal conditions for the long-distance collection, transportation, and dispersion of related mobile genetic elements of microorganisms. Pathogens can invade new locations via MP dispersal. Moreover, natural and non-pathogenic microorganisms increase the chance of acquiring and rapidly spreading AR, which has ultimate adverse effects on several human activities, such as aquaculture resources [77].

Some studies have highlighted the spread of pathogenic bacteria coupled with ARGs through microplastics [80]. For example, *Pseudomonas*, *Aeromonas* spp., *Vibrio* spp*.,* and *E. coli*, known to be opportunistic fish and human pathogens, were found in microplastic biofilms gaining ARGs [51,55]. Moreover, the transfer frequency of plasmids harboring ARGs from *E. coli* to other bacteria in vitro increases when bacteria are associated with MPs. In this context, the spread of ARBs has been found to be associated with MPs in marine ecosystems [77]. In marine aquacultures, there has been an increase in the ARB number on MP surfaces to an order of 100–5000 times more than in the surrounding water, thus also compromising food safety [81]. Moreover, the development rate and chlorophyll content of cyanobacteria *Anabaena* will be considerably lower after the adsorption of macrolide antibiotics on MP [82]. Similarly, the combined presence of tetracycline and polystyrene could worsen the damage caused by oxidation of juvenile *Ctenopharyngodon idella* and intestines and gill tissue injuries [83]. Again, clams could bioaccumulate in the blood increased levels of antibiotics, such as oxytetracycline or florfenicol, if co-exposed with MPs, which further undermined food safety [84]. Similar results have been demonstrated in other environments, such as agricultural soils. Indeed, Wu et al. [85] investigated AR in samples of Chinese soils with long-term exposure to plastic mulch applications and found they had a higher abundance of ARGs, and thousands of mobile genetics elements (MGEs-14 integrons, 28 insertions, and 2993 plasmids). 

#### 1.1.3. Relationship between Heavy Metals, Microplastics and Antibiotic Resistance

Several studies demonstrated both the ubiquity and the negative potential of pollutants such as heavy metals, MPs, and antibiotics on ecosystem safety. Furthermore, since MPs are hydrophobic and represent large attractable areas, hydrophobic pollutants are easily adsorbed on their surfaces, with the consequence of their bioavailability and chemical properties being modified [86,87]. To date, the concentrations of heavy metals and organic pollutants found on the surface of MPs are 10^6^ times higher than the ones present in the close vicinity [19], and this yields combined toxic effects for the surrounding environment: recent studies showed how chemicals, attached, or embedded on MP biofilms, are released because of the natural weathering process. These chemicals include heavy metals, antibiotics, and other xenobiotics. MPs can therefore act as potential carriers of pollution and multidrug resistance in humans [88]. Indeed, there are more metal resistance and multidrug resistance gene types in bacteria isolated from MPs than in free-living strains [89].

The bad implications are related to human health, since these contaminated particles reach humans through ingestion of environmental products. Indeed, humans may feed from seafood such as crustaceans [90], bivalves [91], fish [92], or sea salt [93], which may contain MPs and transmit AR pathogens and metal-driven multi-resistances [24]. Around 80% of fish actually displayed MPs in their stomach [94]; this can cause necrosis of tissues injuries/inflammations or cell necrosis in humans that ingested them [95] or potential cytotoxic complications and oxidative stress on the brain, on epithelial cells, and on the placenta [96,97,98]. In addition, the heavy metals adsorbed on MPs could produce oxygen radicals and damage human cell metabolism [99,100]. 

### 1.2. The AMR Burden: Global Data on Human Health

Although in countries such as the UK and Canada, antibiotic use has been falling since its peak in 2014 (e.g., from 2015 to 2019, in the UK the use has dropped from 19.4 to 17.9 defined daily doses (DDDs) per 1000 inhabitants per day), both hospital and community settings showed again increases in use (+3.5% over the last 5-years in the UK, +30% as antimicrobial purchasing by hospitals and +10% of human consumption in Canada) [101,102]. 

Moreover, to understand the burden of AMR and multidrug resistance pathogens, to foresee the future of human health, and to establish informed surveillance plans, several countries all over the world have launched reports on AMR-related deaths in humans, highlighting dreadful data. In a recent and important paper, Murray et al. overviewed 4.95 million deaths related to bacterial AMR in 204 countries and territories in 2019, of which 1.27 million deaths can be directly attributable to bacterial AMR. ARB caused 929,000 deaths, while 3.57 million deaths were due to AMR-indirect causes. The main ARB species were *Acinetobacter baumannii*, *E. coli*, *Klebsiella pneumoniae*, *Pseudomonas aeruginosa*, *Staphylococcus aureus*, and *Streptococcus pneumoniae*. Furthermore, all-age mortality rates for AMR were highest in some low- and middle-income countries, such as sub-Saharan Africa, raising the problem of AMR as a troublesome issue for some of the poorest countries in the world [103]. 

Between 2014 and 2019, Canada and the UK recorded an increment in the incidence of bloodstream infection with key-bacterial species such as *E. coli*, *K. pneumoniae,* and *Enterococcus* spp. In addition, a doubled rate of healthcare-associated vancomycin-resistant *Enterococcus* has been recorded, and a 140% increase in the rate of community-associated methicillin-resistant *S. aureus*, leading to an estimated 178 new antibiotic-resistant infections per day. The major concern is that around 20% of patients diagnosed with these antimicrobial-resistant bloodstream infections died within 30 days of diagnosis [101,102]. 

These data also parallel those of O’Neill’s report, where 10 million people per year were expected to die because of AMR by 2050 [104]; and this report did not even consider how broad the effect of the COVID-19 pandemic on antibiotic consumption and misuse can be, and therefore AMR spread. Similarly, the Centers for Disease Control and Prevention (CDC) Antibiotic Resistant Threats in the United States 2019 found that 2.8 million resistant infections are responsible for 35,900 deaths annually, with only *Clostridioides difficile* infection killing 12,800 people [105]. Cassini et al. [106] assessed approximately 30,000 deaths and 796,000 disability-adjusted life years caused by AMR in the EU in 2015. Not surprisingly, the Organization for Economic Co-operation and Development (OECD) Health Committee, together with the European Centre for Disease Prevention and Control, projected that in the EU and USA, resistant infections are responsible for about 60,000 deaths a year [105]. Further investigations at the regional level considering patient demographics and healthcare interactions are now being performed to identify better areas of intervention and to develop a more efficient understanding of the AMR scale.

### 1.3. A One Health Approach?

The challenges posed by the above-mentioned stressors concurrently engage human, animal, and environmental health. They are all indeed susceptible to such threats, and their related issues could benefit from a transdisciplinary approach [3]. 

The One Health approach is a concept of “designing and implementing programs, policies, legislation, and research in which multiple sectors communicate and work together to achieve better public health outcomes”. One Health offers a different and multidisciplinary viewpoint, trying to blend data on animal, human, and environmental health. One Health recognizes the interconnection of ecosystem health. The key point of the One Health approach is the continuous dialogue between experts, scientists, and professionals to find potential global health solutions [60,107]. 

Pathogenic ARB dispersal could be managed at multiple levels: improvement of antimicrobial prescriptions; antibiotic policies and legislations, as well as infection prevention and control; integrated surveillance, together with antimicrobial handling, sanitation, and animal husbandry [107], to wastewater treatment or mitigation measures for MPs since they contribute to the rise of AMR.

The management of MPs needs a targeted One Health approach because they can affect multiple ecological compartments, leading to potential ecosystem impacts that ultimately threaten public health [108]. The “access to food for all” (food certainty) would face the threat of MP, with the consequent worsening of human nutritional deficiencies and diseases in some countries [109]. Only a multidisciplinary approach has the potential to be efficacious as a tool for assessing such effects and mitigating them with the intervention of multiple partners with broad expertise. 

## 2. Discussion/Pitfalls

Data obtained from national and international reports show that the burden of AMR and the number of infections caused by pathogens resistant to one or more key antibiotics continue to rise globally. Resistance to multiple agents leads to increased use of last-resort antibiotics, thus enhancing the vicious circle of AMR. Research suggests that explicit awareness and understanding are essential for the distribution and frequency of AMR related to pollutants such as MPs and heavy metals. The microbiological activity involved in some environmental contexts and samples (e.g., agricultural system-irrigation water, compost, manure), as well as correlations between the prevalence of ARB and MPs or metals, is still a huge scientific room to explore.

MPs, heavy metals, antibiotics, resistance genes, and other pollutants ubiquitously present in the environment, and their latent aversive consequences on the biological community are alarming environmental challenges. Heavy metals promote selection or co-selection for ARGs. Metallic NPs have exhibited antimicrobial activity against different microorganisms and Gram-negative and Gram-positive bacteria but also the spreading of AR, at the point that their mechanisms of action are still questioned and remain poorly understood. Since the usage of metallic NPs showed great potential, it is essential to shed light on the genetic mechanisms behind the rise of resistance [37].

Most of the studies covered the investigation of the distribution, abundance, concentration, and toxicological implications of pollutant takeover by organisms. MPs soak up heavy metals and antibiotics, inducing elements and resistance gene transfer and switching the plastisphere as potentially toxic for the surrounding organisms. In this context, the influence of MP biofilms on the spread of ARGs and multi-resistance genes is still too vague. Additionally, little research has taken into account a study about the way to efficiently reduce persistent pollutants such as MPs, heavy metals, antibiotics, and resistance genes in the environment. Moreover, the possibility that the MP surface acts as a binding material for heavy metal ions occurring in the soil system, with unexpected consequences in antibiotic resistance mediated by the heavy metal efflux pump, also deserves focused studies. Likewise, it is still not fully clear how biofilm forms on MPs (e.g., processes and mechanisms behind it). Elucidations on microbe succession and on the different factors affecting biofilm formation on MPs particles, such as the environmental conditions and the age of microplastic particles, are required [110,111]. In addition, how and how much MPs impact the functionality of environmental microbiota is very unclear. Even though the effects of MPs and their involvement in troublesome worldwide issues have been assessed, there is also a need to recognize the lack of scientific maturity in the field [112], which requires growing collaborative efforts from different sectors to address mitigation measures [113,114]. Thus, since MPs are abundant and ubiquity, the risks of their dispersal need to be carefully studied, considering the biotic and abiotic effects on the ecosystem that potentially lead to unrecoverable global changes. Up to now, MP-related research has rarely focused on the ecosystems’ functionality, their abiotic effects, or on the chemical changes in the matrices (sediments, soil aggregates, humus). The ecological significance of their presence in the environment is completely obscure. For instance, the accumulation of MP particles could affect the thermal conductivity or water loss in soils and sediments properties, influencing both the micro- and macrobiota. The risks related to pathogen presence and human or fish exposure, as well as MP-associated pathogens’ capability to cause infectious diseases, are still underestimated, and in-depth risk studies are warranted.

Similarly, the large-scale effects of the accumulation of MPs in several environments have not been sufficiently studied. More research must be financed to support mesocosm studies, long-term experiments, and in-field trials. Although HGT has been shown to increase with MPs [77,115], the MPs’ role in the evolution and dissemination of antibiotic resistance genes in both environmental and pathogen bacteria remains unknown. A further assessment of the indirect effects of MPs in terms of associations and interactions between microplastics and ARGs, especially on the marine environment and sea food safety, is needed. 

In addition, it is important to finance more studies on MPs-associated bacteria whole genomes and their metabolic potential [116,117]. Hence, since only a small portion of environmental bacteria could be cultivated in laboratories, culture-based methods based on bacteria isolation on a culture media, to which antibiotic sensitivity testing is followed, have limited usage in studying antibiotic resistance [118,119,120]. This detection limit can be easily overcome with methods that use total genomic DNA extracted from a given sample, such as quantitative polymerase chain reaction (qPCR) analysis or shotgun metagenomics, which give a general outline of the total bacteria and their associated genes inside the investigated sample [121]. In particular, with tools such as next-generation sequencing technology or whole-genome sequencing (WGS), genotyping pathogens has become very easy [122]. WGS analyses are indeed very efficient in displaying the total metabolic potentiality of microorganisms in order to better understand the basis of antibiotic resistance genetics [123,124]. Despite this, WGS findings on microorganisms related to MP are lacking to a large extent. 

## 3. Conclusions

MPs, antibiotics, and other xenobiotics, such as heavy metals and NPs, are a threat to ecosystems, even due to their chemical properties or to their environmental concentration. These environmental contaminants may stick to the surface of MPs and aggravate the health of the organisms that ingest them. Because of the accumulation of heavy metals at critical concentrations in the environment, bacterial antibiotic resistance may be co-selected while also triggered by co-resistance or cross-resistance. This makes MPs upcoming spots for the co-selection of metal-driven multidrug-resistant pathogens. Additionally, the microbial community in the plastisphere showed a faster HGT rate than the free-living microorganisms, highlighting another cause for environmental and human concerns.

The ultimate consequences for human and animal health are largely to be understood. Their single or correlated effects could affect the microbiota species composition and functionality, shift the physical properties of habitats, modify the nutrient fluxes, and ultimately, the overall ecosystem functions. These accumulative changes may bring to a minor ecosystem resilience, with very serious effects on the Earth’s natural systems. The lack of knowledge regarding it requires new approaches to promoting and assessing interventions. 

The One Health approach could truly create a flow of information between different fields, as environmental, animal, and human health stand in interdependency. The integration of professional figures, even with very different backgrounds, such as biologists, chemists, engineers, health professionals, economists, and policymakers, could help assess direct and indirect adverse effects and propose solutions that can mitigate these threats for the next generation.

Putting this into a scientific perspective, future research directions should focus on (1) the improvement of the MPs monitoring system, innovation, and standardization of research methods; (2) the investigation of the MPs’ long-term implications on the ecology of the environment, and their related acute and chronic toxicity effects; (3) the upgrading of the methods used for heavy metals, antibiotics, pollutant removal, and treatment (e.g., by exploiting submerged plants to absorb MPs in waters); (4) on the study of MPs degradation processes, that are still scarcely investigated; and (5) on the assessment and quantification of the frequency of AR-HGT on MP biofilms and on the comparison of these results with natural AR-HGT, both in-vitro studies and in natural environmental matrices.

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
