# Peer review of "Understanding the Interplay between Antimicrobial Resistance, Microplastics and Xenobiotic Contaminants: A Leap towards One Health?"

_ijerph, 2022, doi:10.3390/ijerph20010042_

Round 1

Reviewer 1 Report

   The authors present shortly a state of knowledge in the area of interaction between microplastics, xenobiotic contaminants and antibiotic resistance. They quote in the Introduction a number of studies in this field.

   However, the discussion in the next part of the manuscript is strongly too brief and vague. This part of the submission should be firmly enhanced.

   I suggest also a few specific amendments as follows:

 - The language should be improved, especially with regard of the syntax and wording. The whole text should be revised in this respect.

 - The keywords ‘environment’ and ‘global health’ are too general, overly broad. I suggest indicate more specific keywords instead.

 - Abstract – named ‘Astract’ (typo).

 - The page 2 – repetition ‘their pervasive’.

 - The page 3 – it should be rather ‘100-5000 times more’ (or 2-4 orders of magnitude), not ‘an order of 100-5000 magnitudes’?. It should be corrected.

 - The page 3 – redundant space: 140 %.

 - The authors quote in the Introduction numerous articles in the field but in the Discussion, as I said earlier, too vaguely, superficially and too briefly discuss identified topics. The discussion must be extended and detailed. In present form is very shallow.

 - The Conclusions – also too vague and superficial, too general. Must be revised to make it more detailed and referred precisely to identified earlier in the previous sections indications.

Author Response

The authors present shortly a state of knowledge in the area of interaction between microplastics, xenobiotic contaminants and antibiotic resistance. They quote in the Introduction a number of studies in this field.

However, the discussion in the next part of the manuscript is strongly too brief and vague. This part of the submission should be firmly enhanced.

I suggest also a few specific amendments as follows:

 - The language should be improved, especially with regard of the syntax and wording. The whole text should be revised in this respect.

The revised version has been checked regarding the English style.

 - The keywords ‘environment’ and ‘global health’ are too general, overly broad. I suggest indicating more specific keywords instead.

We improved the keyword in a more specific way.

 - Abstract – named ‘Astract’ (typo).

Done

 - The page 2 – repetition ‘their pervasive’.

Done

 - The page 3 – it should be rather ‘100-5000 times more’ (or 2-4 orders of magnitude), not ‘an order of 100-5000 magnitudes’?. It should be corrected.

Done

 - The page 3 – redundant space: 140 %.

Done

 - The authors quote in the Introduction numerous articles in the field but in the Discussion, as I said earlier, too vaguely, superficially and too briefly discuss identified topics. The discussion must be extended and detailed. In present form is very shallow.

WE rewrite all the parts of the article that were very general, including much more literature and deepen the discussion as requested.

 - The Conclusions – also too vague and superficial, too general. Must be revised to make it more detailed and referred precisely to identified earlier in the previous sections indications.

We went on the discussion and perspectives, detailing them better and recapping the concepts highlighted in previous sections.

Reviewer 2 Report

My comments and questions are:

Important Note: I have checked the manuscript and, with the assistance of originality checking software, namely iThenticate®-Professional Plagiarism Prevention. 24% similarity appears when I checked the whole manuscript PDF (excluding References section).

11)      There are typographical errors in the Text and References. Generally, all manuscript sections (in-text citations and reference) should be reviewed according to reference citation guide of the JERPH.  For examples

·         in the page 1: “(Astract)” Should be changed as “(Abstract)”.

22)      The second sentence od the abstract is too long for clear undertanding. This sentence should be re-organise as two sentences.

33)      Please indicate the long form of AMR in Abstract section.

44)      A sentence about the combined effects of microplastics and antibiotics should be added to the abstract section.

55)      Please indicate the long form of AMR in Keywords section.

66)      New words should be added as keywords. For example; Xenobiotic Contaminants.

77)      New dated references should be cited in introduction in terms of the presence of significant concentration of antibiotics [1,2]

88)      A new subsection should be added related with the relationship between nanomaterials which is another important class of environmental pollutants and antimicrobial resistance. There are a few examples article about this topic:

·         Miller, J. H., Novak, J. T., Knocke, W. R., Young, K., Hong, Y., Vikesland, P. J., ... & Pruden, A. (2013). Effect of silver nanoparticles and antibiotics on antibiotic resistance genes in anaerobic digestion. Water environment research, 85(5), 411-421.

·         Amaro, F., Morón, Á., Díaz, S., Martín-González, A., & Gutiérrez, J. C. (2021). Metallic nanoparticles—friends or foes in the battle against antibiotic-resistant bacteria?. Microorganisms, 9(2), 364.

99)      Further publications should be discussed to indicate the relationship between microplastics, antibiotic resistance and human health:

·         Marathe, N. P., & Bank, M. S. (2022). The Microplastic-Antibiotic Resistance Connection. In Microplastic in the Environment: Pattern and Process (pp. 311-322). Springer, Cham.

·         Pham, D. N., Clark, L., & Li, M. (2021). Microplastics as hubs enriching antibiotic-resistant bacteria and pathogens in municipal activated sludge. Journal of Hazardous Materials Letters, 2, 100014.

110)  Please try to concentrate on a single point in each paragraph. Tie together like paragraphs with a conclusion paragraph. Deepen the contents of paragraphs. Don't simply list studies.

111)  Possible suggestions for future perspective can be presented.

Author Response

Important Note: I have checked the manuscript and, with the assistance of originality checking software, namely iThenticate®-Professional Plagiarism Prevention. 24% similarity appears when I checked the whole manuscript PDF (excluding References section).

Some citations or quotations cannot be changed: the most important thing is to insert the reference where they come from.

1)      There are typographical errors in the Text and References. Generally, all manuscript sections (in-text citations and reference) should be reviewed according to reference citation guide of the JERPH.  For examples: in the page 1: “(Astract)” Should be changed as “(Abstract)”.

Done

2)      The second sentence od the abstract is too long for clear undertanding. This sentence should be re-organise as two sentences.

The reviewer has right. We split it in more sentences.

3)      Please indicate the long form of AMR in Abstract section.

Done

4)      A sentence about the combined effects of microplastics and antibiotics should be added to the abstract section.

We added a sentence as requested.

5)      Please indicate the long form of AMR in Keywords section.

Done

6)      New words should be added as keywords. For example; Xenobiotic Contaminants.

Done

7)      New dated references should be cited in introduction in terms of the presence of significant concentration of antibiotics [1,2]

We updated the references.

8)      A new subsection should be added related with the relationship between nanomaterials which is another important class of environmental pollutants and antimicrobial resistance. There are a few examples article about this topic:

-         Miller, J. H., Novak, J. T., Knocke, W. R., Young, K., Hong, Y., Vikesland, P. J., ... & Pruden, A. (2013). Effect of silver nanoparticles and antibiotics on antibiotic resistance genes in anaerobic digestion. Water environment research, 85(5), 411-421.

-         Amaro, F., Morón, Á., Díaz, S., Martín-González, A., & Gutiérrez, J. C. (2021). Metallic nanoparticles—friends or foes in the battle against antibiotic-resistant bacteria?. Microorganisms, 9(2), 364.

We changed as requested.

9)      Further publications should be discussed to indicate the relationship between microplastics, antibiotic resistance and human health:

-         Marathe, N. P., & Bank, M. S. (2022). The Microplastic-Antibiotic Resistance Connection. In Microplastic in the Environment: Pattern and Process (pp. 311-322). Springer, Cham.

-         Pham, D. N., Clark, L., & Li, M. (2021). Microplastics as hubs enriching antibiotic-resistant bacteria and pathogens in municipal activated sludge. Journal of Hazardous Materials Letters, 2, 100014.

We added the two suggested articles and deepened better the section.

10)  Please try to concentrate on a single point in each paragraph. Tie together like paragraphs with a conclusion paragraph. Deepen the contents of paragraphs. Don't simply list studies.

We revised the text according to the suggestion, especially the old 1.1 paragraph, that had several citations without a real deepening of their message. That was mostly due to the manuscript’s size limits in term of number of characters. However, due to this important request, we decided to put more information to meet criticism.

11)  Possible suggestions for future perspective can be presented.

We rewrote the perspective paragraph including possible views for the future research.

Reviewer 3 Report

This perspective presents the relationship between antimicrobial resistance, microplastics and xenobiotic contaminants. Authors give a brief insight on the antimicrobial resistance and

 antibiotic-resistant genes related to urban discharge into inland waters, metal and microplastics contamination and their impact on ecosystems and human health. Then authors highlight the need of one health approach to counter these problems. The work is well designed and provides information worth knowing

Minor points:

Repetitions in the sentence: “bonds, their pervasive their pervasive presence in all the environments”

The sentence “WWTPs have been not design to remove microplastics” to be replaced by “WWTPs have been not designed to remove microplastics”

Author Response

This perspective presents the relationship between antimicrobial resistance, microplastics and xenobiotic contaminants. Authors give a brief insight on the antimicrobial resistance and antibiotic-resistant genes related to urban discharge into inland waters, metal and microplastics contamination and their impact on ecosystems and human health. Then authors highlight the need of one health approach to counter these problems. The work is well designed and provides information worth knowing

Minor points:

Repetitions in the sentence: “bonds, their pervasive their pervasive presence in all the environments”

Done

The sentence “WWTPs have been not design to remove microplastics” to be replaced by “WWTPs have been not designed to remove microplastics”

Done

Round 2

Reviewer 2 Report

The authors performed the required modifications, according to the suggestions. Revised form of the manuscript can be accepted.